# Cytotoxic T Lymphocytes Control Growth of B16 Tumor Cells in Collagen–Fibrin Gels by Cytolytic and Non-Lytic Mechanisms

**DOI:** 10.3390/v15071454

**Published:** 2023-06-27

**Authors:** Barun Majumder, Sadna Budhu, Vitaly V. Ganusov

**Affiliations:** 1Department of Microbiology, University of Tennessee, Knoxville, TN 37996, USA; 2Department of Pharmacology, Weill Cornell Medicine, New York, NY 10021, USA; sab4028@med.cornell.edu; 3Department of Mathematics, University of Tennessee, Knoxville, TN 37996, USA

**Keywords:** cytotoxic T lymphocytes, killing, B16 tumors, mathematical modeling

## Abstract

Cytotoxic T lymphocytes (CTLs) are important in controlling some viral infections, and therapies involving the transfer of large numbers of cancer-specific CTLs have been successfully used to treat several types of cancers in humans. While the molecular mechanisms of how CTLs kill their targets are relatively well understood, we still lack a solid quantitative understanding of the kinetics and efficiency by which CTLs kill their targets in vivo. Collagen–fibrin-gel-based assays provide a tissue-like environment for the migration of CTLs, making them an attractive system to study T cell cytotoxicity in in vivo-like conditions. Budhu.et al. systematically varied the number of peptide (SIINFEKL)-pulsed B16 melanoma cells and SIINFEKL-specific CTLs (OT-1) and measured the remaining targets at different times after target and CTL co-inoculation into collagen–fibrin gels. The authors proposed that their data were consistent with a simple model in which tumors grow exponentially and are killed by CTLs at a per capita rate proportional to the CTL density in the gel. By fitting several alternative mathematical models to these data, we found that this simple “exponential-growth-mass-action-killing” model did not precisely describe the data. However, determining the best-fit model proved difficult because the best-performing model was dependent on the specific dataset chosen for the analysis. When considering all data that include biologically realistic CTL concentrations ($E\leq 10^{7}\mathrm{cell}/\mathrm{mL}$), the model in which tumors grow exponentially and CTLs suppress tumor’s growth non-lytically and kill tumors according to the mass–action law (SiGMA model) fit the data with the best quality. A novel power analysis suggested that longer experiments (∼3–4 days) with four measurements of B16 tumor cell concentrations for a range of CTL concentrations would best allow discriminating between alternative models. Taken together, our results suggested that the interactions between tumors and CTLs in collagen–fibrin gels are more complex than a simple exponential-growth-mass–action killing model and provide support for the hypothesis that CTLs’ impact on tumors may go beyond direct cytotoxicity.

## 1. Introduction

Cytotoxic T lymphocytes (**CTLs**) are important in controlling viral infections and tumors [1,2]. CTLs exhibit such control via several complimentary mechanisms including direct cytotoxicity—the ability of CTLs to kill virus-infected or tumor (target) cells. Killing of a target cell by a CTL in vivo is a multi-step process: (1) CTLs must migrate to the site where the target is located; (2) CTLs must recognize the target (typically by the T cell receptor (**TCR**) on the surface of T cells binding to the specific antigen presented on the surface of the target cell); (3) CTLs must form a cytotoxic synapse with the target; (4) CTLs must induce the apoptosis of the target cell by secreting effector molecules (e.g., perforin and granzymes) or through Fas/Fas-ligand interactions [3,4,5,6,7]. The relative contribution of these steps to the efficiency at which a population of CTLs kill their targets in vivo remains poorly understood especially in complex tissues. Improving the efficacy of cancer immunotherapies such as adoptive transfer of cancer-specific T cells will likely come from better understanding of a relative contribution of these processes to tumor control [8].

Many previous studies have provided quantitative insights into how CTLs eliminate their targets in vitro. The first insights came from generating conjugates between target cells and CTLs and quantifying how quickly a target cell dies when either being bound by a different number of CTLs or when one CTL binds to different targets [9,10,11,12,13,14,15,16]. Further in vitro studies highlighted that killing by CTLs may kill multiple targets rapidly [17,18,19], but also highlighted heterogeneity in the efficacy at which individual CTLs kill their targets [20,21]. Interestingly, killing of tumor cells in vitro may take a long time (hours) with speed and turning being important in determining the likelihood that a CTL will find and kill the target [22,23]. One study suggested that killing of targets in vitro may follow the law of mass–action [24]. The killing efficiency of CTLs has also been evaluated in so-called chromium release assays, which have been a standard method in immunology to measure T cell cytotoxicity in vitro [25,26,27,28,29,30,31,32].

Evaluating the killing efficacy of CTLs in vivo is challenging. One approach to evaluate how a population of CTLs eliminates targets in vivo has been to perform in vivo cytotoxicity assays [33]. In the assay, two populations of cells, pulsed with a specific peptide and another being a control, are transferred into mice carrying peptide-specific CTLs, and a relative percent of peptide-pulsed targets is determined in a given tissue (typically spleen) after different times after the target cells are transferred [33,34,35]. Different mathematical models have been developed to determine specific terms describing how CTLs kill their targets and to estimate CTL killing efficacy; such estimates varied orders of magnitude between different studies often using similar or even the same data [36,37,38,39,40,41,42,43]. One study suggested that a mass–action killing term is fully consistent with the data from different in vivo cytotoxicity experiments [41], while other studies based on theoretical arguments suggested that killing should saturate at high CTL or target cell densities [37,44,45].

Intravital imaging has provided additional insights into how CTLs kill their targets in vivo [46,47]. One pioneering study followed interactions between peptide-pulsed B cells and peptide-specific CTLs in lymph nodes of mice and found that CTLs and their targets form stable conjugates and move together until the target stops and dies, presumably due to a lethal hit delivered by the CTL [48]. This and other studies revealed that, to kill a target in vivo, CTLs either need to interact with the target for a long time (tens of minutes to hours) or multiple CTLs must contact a target to ensure its death [2,49,50,51,52,53,54]. By measuring the decay of the fluorescence of Plasmodium yoelii sporozoites inside of hepatocytes in murine livers in vivo, we recently showed that the killing of the infected hepatocytes by individual CTLs takes several hours [55]. The time it takes CTLs to kill tumors or Plasmodium-infected hepatocytes is much longer than the half-life of peptide-pulsed targets estimated from in vivo cytotoxicity assays in mice [39,50,51,55]. This may be due to different levels of presented antigens (targets pulsed with a high concentration of a cognate peptide vs. targets expressing endogenous antigens), but may also be due to differences in the intrinsic killing abilities of different T cells. Mathematical modeling provided quantification of how CTLs kill their targets and of various artifacts arising in intravital imaging experiments; for example, one recent study showed how allowing for the retention of CTLs at a dying (or already dead) target cell called “zombie contacts” influences estimates of CTL killing efficacy [56,57]. We recently suggested that the median killing efficacy of individual Plasmodium-specific CTLs is too low to rapidly eliminate a Plasmodium liver stage, highlighting the importance of clusters of CTLs around the the liver stage for its efficient elimination [55].

Even though studying how CTLs kill their targets in vivo is most relevant, such experiments are expensive and time-consuming and have a low throughput. On the other hand, traditional in vitro experiments (e.g., on plates or in wells) suffer from the limitation that CTLs and targets do not efficiently migrate on flat surfaces as they do in vivo in many tissues. Collagen–fibrin gels have been proposed as a useful in vitro system to study CTL and target cell interactions, which allows better representing the complex 3D environment of the tissues with a low cost and higher throughput [58,59,60]. CTLs readily migrate in these gels with speeds similar to that of T cells in some tissues in vivo [61,62]. One recent study measured how CTLs derived from transgenic mice whose TCRs are all specific for the peptide SIINFEKL (from chicken ovalbumin) can eliminate SIINFEKL-pulsed B16 tumor cells in collagen–fibrin gels [58]. Interestingly, the rate at which tumor cells were lost from the gel was linearly dependent on the concentration of CTLs in the gel (varying from 0 to $10^{7}$ cell/mL) and was independent of the number of B16 tumor cells deposited in the gel [58]. This result suggested that the killing of B16 tumor cells in collagen–fibrin gels follows the law of mass–action, and given that the population of B16 tumor cells grew exponentially with time in the absence of CTLs, the authors proposed that $3.5\times 10^{5}\;\mathrm{cell}/\mathrm{mL}$ of CTLs are required to prevent B16 tumor cell accumulation in these gels.

In the present paper, by using mathematical modeling, we re-analyzed the data of Budhu et al. [58] including two additional previously unpublished datasets on the CTL killing of B16 tumor cells in collagen–fibrin gels. We found that the simple exponential growth and mass–action killing model never provided the the best fit of the data. The best-fit model in the list of the four tested model was dependent on a specific dataset used for the analysis. The model in which CTLs reduced the growth rate of B16 tumor cells and killed the tumors via a mass–action law (proportional to concentrations of the CTLs and tumors) fit our largest dataset consisting of 431 gels the best. Importantly, the type of the model was critical in predicting the CTL concentration that would be needed to eliminate most of the tumor cells within a defined time period (e.g., 100 days), suggesting the need for future experiments to discriminate between alternative models. Following our recent framework for experimental power analyses [63], we simulated three different experimental designs and found which designs would allow better discriminating between alternative mathematical models of the CTL-mediated control of B16 tumor cells and, thus, will allow better predicting how many CTLs are needed for tumor control.

## 2. Materials and Methods

### 2.1. Experimental Details and Data

The details of the experimental designs were described previously [58]. In short, $10^{3}$–$10^{6}$ SIINFEKL-pulsed B16 melanoma tumor cells (=$10^{4}$–$10^{7}$ cell/mL) were inoculated alone or with $10^{3}$–$10^{6}$ (equivalent to $10^{4}$–$10^{7}$ cell/mL) of activated OT1 T cells (CTLs) into individual wells containing collagen–fibrin gels of a total volume of 0.1 mL. At different times after co-inoculation of cells, gels were digested, and the resulting solution was diluted $10^{1}$–$10^{3}$-fold (depending on the initial desired B16 cell concentration) in growth medium; the number of surviving B16 cells in each gel was counted [58]. The data represent the concentration of B16 tumor cells (in cell/mL) surviving in the gels for a given time. Budhu et al. [58] kindly provided us with the data from their published experiments (Datasets 1, 2, and 3), as well as two additional unpublished datasets (Datasets 4 and 5). A summary of these datasets is as follows:**Dataset 1** (“growth”): SIINFEKL-pulsed B16 melanoma cells were inoculated in 3D collagen-I–fibrin gels with the desired initial concentrations of $10^{3}$, $10^{4}$, or $10^{5}$ cell/mL and no OT1 cells. The surviving B16 cells were measured at 0, 24, 48, and 72 h after inoculation into gels. The total number of data points (gels) n=70 (Figure A1A).**Dataset 2** (“short-term growth and killing”): SIINFEKL-pulsed B16 melanoma cells were inoculated with the desired initial concentrations of $10^{4}$, $10^{5}$, or $10^{6}$ cell/mL with activated CD8${}^{+}$ OT1 cells with concentrations of 0, $10^{4}$, $10^{5}$, $10^{6}$, or $10^{7}$ cell/mL. The surviving B16 cell numbers were measured at 0 and 24 h. The total number of data points n=175 (Figure A1B).**Dataset 3** (“long-term growth and killing”): SIINFEKL-pulsed B16 melanoma cells were inoculated with the desired initial concentrations of $10^{6}$ or $10^{8}$ cell/mL with OT1 T cells with concentrations of 0, $10^{6}$, or $10^{7}$ cell/mL. Gels with a B16 cell concentration of $10^{8}$ cell/mL were unstable and, thus, were not included in the analysis. The measurements of surviving B16 cells were performed at at 0, 24, 48, 72, and 96 h post-inoculation into the gels. The total number of data points n=96 (Figure A1C).**Dataset 4** (“growth and killing in the first 24 h”): In this previously unpublished dataset, SIINFEKL-pulsed B16 melanoma cells were co-inoculated into gels with the desired initial concentration of $10^{5}$ cell/mL and with OT1 T cells at concentrations of 0, $10^{6}$, or $10^{7}$ cell/mL. Surviving B16 cells were measured at 0, 4, 8, 12, and 24 h post-inoculation into the gels. The total number of data points n=90 (Figure A1D).**Dataset 5** (“killing at a high CTL concentration”): In this previously unpublished dataset, SIINFEKL-pulsed B16 melanoma cells were co-inoculated into gels at the desired initial concentration of $10^{5}$ cell/mL and with OT1 cells at concentrations of 0 or $10^{8}$ cell/mL. Surviving B16 cells were measured at 0 and 24 h. The total number of data points n=7 (Figure A1E).

Experiments generating the data for Datasets 1–4 were repeated three times (Experiments 1, 2, and 3), and each measurement was performed in duplicate [58]. These experimental duplicates were prepared for each experimental condition; at a time point, each of the two gels was lysed and diluted, and the cells from each gel were plated into two $65\times 15\;{\mathrm{mm}}^{2}$ plates. The experiment generating Dataset 5 was performed once.

### 2.2. Mathematical Models

#### 2.2.1. Mathematical Models to Explain Tumor Dynamics

Given previous observations of Budhu et al. [58], we assumed that B16 melanoma (tumor) cells grow exponentially and are killed by OT1 CD8${}^{+}$ T cells (CTLs) at a rate proportional to the density of the tumors. The change in the B16 cell concentration (*T*) over time is then described by a differential equation of the general form
(1)$$\frac{dT}{dt}=f_{g}\left(E\right)T-f_{k}\left(E\right)T,$$
where $f_{g}\left(E\right)$ is the per capita growth rate, $f_{k}\left(E\right)$ is the death rate of tumors, and *E* is the CTL concentration. When *E* is constant, the general solution of this equation can be written as
(2)$$\ln T\left(t\right)=\ln\left(\frac{T_{a}}{\alpha }\right)+\left(f_{g}\left(E\right)-f_{k}\left(E\right)\right)t,$$
where we let $T\left(0\right)=T_{a}/\alpha$ be the initial count, which depends on the desired B16 tumor cell concentration $T_{a}$ subject to an effective fraction 1/α of cells. It was typical to recover a somewhat lower B16 cell number from the gel than when it was inoculated. For example, when aiming for $1\times 10^{4}$ B16 tumor cells per mL in a gel, it was typical to recover ∼0.4 $\times 10^{4}$ cell/mL at Time 0 (e.g., Figure A1A).

The simplest exponential growth and mass–action killing (**MA**) model assumes that tumors grow exponentially and are killed by CTLs at the rate proportional to the CTL density ($f_{g}\left(E\right)=r$ and $f_{k}\left(E\right)=kE$, Figure 1A). Using Equation (Equation 2), the change in the density of targets over time is given by
(3)$$\ln T\left(t\right)=\ln\left(\frac{T_{a}}{\alpha }\right)+\left(r-kE\right)t.$$

This model has three parameters (*r*, *k*, and α) to be estimated from the data.

The second “saturation” (**Sat**) model assumes that tumors grow exponentially and are killed by CTLs at a rate that saturates at high CTL densities ($f_{g}\left(E\right)=r$ and $f_{k}\left(E\right)=\frac{kE}{h+E}$, Figure 1B). Using Equation (Equation 2), its solution is
(4)$$\ln T\left(t\right)=\ln\left(\frac{T_{a}}{\alpha }\right)+\left(r-\frac{kE}{h+E}\right)t.$$

This model has 4 parameters (*r*, *k*, *h*, and α) to be estimated from the data.

The third “**Power**” model assumes that grow exponentially and are killed by CTLs at a rate that scales as a power law with CTL density ($f_{g}\left(E\right)=r$ and $f_{k}\left(E\right)=kE{}^{n}$, Figure 1C). Using Equation (Equation 2) its solution is
(5)$$\ln T\left(t\right)=\ln\left(\frac{T_{a}}{\alpha }\right)+\left(r-kE^{n}\right)t.$$

This model has 4 parameters (*r*, *k*, *n*, and α) to be estimated from the data.

In the fourth suppression-in-growth with mass–action-killing (**SiGMA**) model, we assumed that CTLs suppress the growth rate of the tumor and kill the tumor according to the mass–action law ($f_{g}\left(E\right)=g_{0}+\frac{g_{1}}{1+E/g_{2}}$ and $f_{k}\left(E\right)=kE$, Figure 1D). Using Equation (Equation 2), its solution is
(6)$$\ln T\left(t\right)=\ln\left(\frac{T_{a}}{\alpha }\right)+\left(g_{0}+\frac{g_{1}}{1+E/g_{2}}-kE\right)t,$$
where $g_{0}$ is the B16 tumor growth rate, which is independent of the CTLs, $g_{1}$ the tumor growth rate that can be reduced by CTLs via non-lytic means, and $g_{2}$ the density of CTLs at which the growth rate $g_{2}$ is reduced to half of its maximal value due to CTL activity. Note that, in this model, the rate of tumor cell replication in the absence of CTLs is $r=g_{0}+g_{1}$. This model has 5 parameters ($g_{0}$, $g_{1}$, $g_{2},k$, and α) to be estimated from the data.

#### 2.2.2. Estimating Initial Density of Tumor Cells in Gels

In the general solution (Equation (Equation 2)), we assumed that the initial tumor density is proportional to the density desired in the experiments scaled by a factor α. We found that recovered concentrations of B16 tumor cells from gels at time t=0 were consistently lower than the desired value, and such reduction was approximately similar for different initial B16 concentrations (results not shown). Experimentally, this may arise because the clonogenic assay used to count the number of B16 tumor cells in the gels was not 100% efficient. To check this assuming an identical effective fraction 1/α for the initial B16 concentration in different experiments, we tested an alternative model, where we assumed different α for different desired B16 cell concentrations. In this varying α model, the first term in Equation (Equation 2) can be written as $\ln\left(\frac{T_{a_{i}}}{\alpha_{i}}\right)$, where *i* denotes the desired B16 cell concentration. For example, a dataset containing the desired B16 concentrations of $10^{5},10^{6}$ and $10^{7}$ cell/mL would have three α: $\alpha_{1}$, $\alpha_{2}$, and $\alpha_{3}$, respectively. To fit the model with dataset-dependent α, we used the function MultiNonlinearModelFit in Mathematica. We found that allowing α to vary between different desired B16 concentrations when fitting the SiGMA model to Datasets 1–4 marginally improved the model fit ($\chi_{4}^{2}=9.5$, p=0.02), but did not influence the estimates of other parameters (Table A2); in our following analyses, we, therefore, opted for the simpler model with a single effective fraction parameter 1/α.

#### 2.2.3. Time to Kill 90% of Targets

To evaluate the efficacy of the CTL-mediated control of tumors, we calculated the time it takes to kill 90% of tumors initially present. For every model (Equations (Equation 3)–(Equation 6)), we solved an equation $f_{g}\left(E\right)t_{90}-f_{k}\left(E\right)t_{90}=\ln\left(0.1\right)$ to find time $t_{90}$ in terms of CTL concentration *E*:(7)$$t_{90}\left(E\right)=\frac{\ln(10)}{f_{k}\left(E\right)-f_{g}\left(E\right)}.$$

#### 2.2.4. Models to Explain Tumor Growth in the First 24 h after Inoculation into Gels

In new experiments (Dataset 4), we found that the growth of the tumors in the first 24 h after inoculation into the gels may not follow a simple exponential curve. Experimentally, this delay may be due to the tumor cells adjusting to the gel environment. In order to explain this dynamics, we propose two additional models. As a first alternative (**Alt 1**) model, we allowed for a natural death of B16 tumor cells, and then, after a delay, growth starts. The motivation for this new growth function came from an algebraic sigmoid function, which changes sign from a constant negative value to a constant positive value. The change in the concentration of B16 tumor cells in the absence of CTLs is given by
(8)$$\ln T\left(t\right)=\ln\left(\frac{T_{a}}{\alpha }\right)+r\sqrt{1+{\left(t-t^{\prime }\right)}^{2}},$$
where the constant $t^{\prime }$ quantifies the time at which this change in sign happens. This model has 3 parameters (α, *r*, and $t^{\prime }$) to be estimated from the data.

As the second alternative (**Alt 2**) model, we considered a more mechanistic explanation of the nonlinear dynamics of the tumor cells. We assumed that a fraction $f_{d}$ of B16 tumor cells die at rate *d* and the rest ($1-f_{d}$) grow at rate *r*. The model can be described by the following equations:(9)$$T\left(t\right)=f_{d}\left(\frac{T_{a}}{\alpha }\right)e^{-dt}+\left(1-f_{d}\right)\left(\frac{T_{a}}{\alpha }\right)e^{rt},$$
where *d* is the death rate of the $f_{d}$ subset of tumor cells. This model has 4 parameters (α, $f_{d}$, *d*, and *r*) to be estimated from the data.

#### Statistics

Natural log-transformed solutions of the models were fit to the natural log of the measured concentrations of B16 tumor cells using least squares. In the data, there were 13 gels (out of 451) that had 0 B16 tumor cells recovered; these data were excluded from most of the analyses. The regression analyses were performed using function NonlinearModelFit in Mathematica (ver 11.3.0.0). For every model, we calculated AIC and Δ as
(10)$$\Delta_{i}={\mathrm{AIC}}_{i}-\min\left\{{\mathrm{AIC}}_{i}\right\},$$
where subscript *i* denotes the model and “min” denotes the minimal AIC for all models [64]. The Akaike weight for the model *i* was calculated as
(11)$$w_{i}=\frac{e^{-\Delta_{i}/2}}{\sum_{i}e^{-\Delta_{i}/2}}.$$

To evaluate the appropriateness of the assumptions of the least-squares-based regressions, we analyzed the residuals of the best fits by visual inspection and using the Shapiro–Wilk normality test using the function ShapiroWilkTest in Mathematica.

## 3. Results

### 3.1. Experiments to Measure How CTLs Kill Targets in Collagen–Fibrin Gels

To estimate the density of the tumor-specific CTLs needed to control the growth of B16 tumor cells, Budhu et al. [58] performed a series of experiments in which variable numbers of SIINFEKL peptide-pulsed B16 tumor cells and SIINFEKL-specific CTLs (activated OT1 CD8 T cells) were co-inoculated in collagen–fibrin gels, and the number of surviving tumor cells was calculated at different time points (see Figure A1 and the Materials and Methods Section for more details). In the absence of CTLs (**Dataset 1**), B16 tumor cells grew exponentially with the growth rate being approximately independent of the initial tumor density (Figure A1A). Short-term (24 h) experiments (**Dataset 2**) showed that, when the density of CTLs exceeded $10^{6}\mathrm{cell}/\mathrm{mL}$, the density of B16 cells declined in 24 h, suggesting that the killing rate of the tumors exceeded their replication rate (Figure A1B). Longer (96 h) experiments (**Dataset 3**) showed that, at high CTL densities (>$10^{6}$ cell/mL), the number of B16 targets recovered from the gels declined approximately exponentially with time; interestingly, however, at an intermediate density of CTLs and B16 tumor cells of $10^{6}\mathrm{cell}/\mathrm{mL}$, the B16 cells initially declined, but then rebounded and accumulated (Figure A1C). Previously unpublished experiments (**Datasets 4–5**) showed a similar impact of increasing CTL density on the B16 tumor dynamics during short-term (24 h) experiments (Figure A1D,E). Budhu et al. [58] concluded that the data from short- and long-term experiments (Figure A1A–C) are consistent with the model in which the number of B16 tumor cells grows exponentially due to cell division and are killed by CTLs at a mass–action rate (proportional to the density of targets and CTLs). Budhu et al. [58] also concluded that a density of 3.5 ×$10^{5}$ cell/mL was critical for the controlling growth of B16 tumor cells in collagen–fibrin gels.

### 3.2. Alternative Models of Growth and Killing May Better Explain the Data Than a Simple Exponential-Growth-and-Mass–Action-Killing Model

The conclusion that a simple model with exponentially growing tumors and killing of the tumors by CTL via the mass–action law (**MA** model, Figure 1A) was based on simple regression analyses of individual datasets (e.g., Dataset 1 or 3). To more rigorously investigate this issue, we propose three additional models, which make different assumptions of how CTLs impact B16 tumor cells including (i) saturation in the killing rate (**Sat** model, Equation (Equation 4) and Figure 1B), (ii) nonlinear change in the death rate of tumors with increasing CTL concentrations (**Power** model, Equation (Equation 5) and Figure 1C), and (iii) reduction in the tumor growth rate with increasing CTL concentrations and the mass–action killing term (**SiGMA** model, Equation (Equation 6) and Figure 1D). We then fit these models including the MA model to all the data, which in total included 438 measurements (and excluded 13 gels with zero B16 tumor cells; see the Materials and Methods Section for more details). These data included two new unpublished datasets (Dataset 4 and 5) including the B16 tumor dynamics at unphysiologically high CTL concentrations ($E=10^{8}\;\mathrm{cell}/\mathrm{mL}$, Figure A1E). Interestingly, we found that the MA model fit these data with the least accuracy while the Sat model (with a saturated killing rate) fit the data best (Table A1). Saturation in the killing rate by CTLs is perhaps not surprising in the full dataset given that, in Dataset 5, two gels inoculated with $10^{5}$ B16 tumor cells and $10^{8}\;\mathrm{cell}/\mathrm{mL}$ CTLs still contained B16 tumor cells at 24 h (Figure A1E). Because $10^{8}\;\mathrm{cell}/\mathrm{mL}$ is a physiologically unrealistic density of CTLs in vivo, for most of our following analyses, we excluded Dataset 5.

Importantly, the MA model was still the least-accurate at describing the data from Datasets 1–4, which is visually clear from the model fits of the data, as well as from the statistical comparison of alternative models using the AIC (Figure 2 and Table 1). In contrast, the SiGMA model provided the best fit (Table 1). The SiGMA model is unique because it suggests that, in these experiments, CTLs impact tumor accumulation not only by killing the tumors, but also by slowing down the tumor rate of growth from the maximal value of $r=g_{0}+g_{1}=0.76/\mathrm{day}$ to the minimal $g_{0}=0.12/\mathrm{day}$ already at moderate CTL concentrations ($E\approx 10^{4}\;\mathrm{cell}/\mathrm{mL}$, Table 1). It is well recognized that CTLs are able to produce large amounts of interferon-gamma (**IFNg**), which may directly inhibit tumor growth, especially of IFNg-receptor-expressing cells [65,66,67]. Interestingly, while statistically, the Sat and Power models fit the data worse than the SiGMA model (Table 1), visually, the fits of these three models were very similar (Figure 2). Furthermore, at high CTL concentrations ($E=10^{7}\;\mathrm{cell}/\mathrm{mL}$), all four models provided fits of a similar quality (Figure 2E).

It is important to note that, even the best-fit SiGMA model did not accurately describe all the data. For example, the model over-predicted the B16 counts at 24 h for OT1 concentrations of $10^{4}$ and $10^{5}$ cell/mL (Figure 2B,C) and under-predicted the B16 counts at 96 h in growth (Figure 2A) and at 72 h for OT1 concentrations $10^{7}$ cell/mL (Figure 2E).

To intuitively understand why the MA model did not fit the data well, we performed several regression analyses. Specifically, for every CTL and B16 tumor cell concentrations, we calculated the net growth rate of the tumors $r_{\mathrm{net}}$ (Figure A1); in cases of several different desired B16 concentrations, we calculated the average net growth rate. In the absence of CTLs, the net growth rate of tumor cells was $r_{0}=0.62/\mathrm{day}$ (Figure A2). Then, for every CTL concentration, we calculated the death rate of B16 tumor cells due to CTL killing as $K=r_{0}-r_{\mathrm{net}}$. For the MA model, the death rate *K* should scale linearly with the CTL concentration [41]; however, we found that this was not the case for B16 tumor cells in gels where the death rate scaled sublinearly with the CTL concentration (Figure A2). Importantly, this analysis also illustrated that, at low CTL concentrations ($10^{4}$–$10^{5}$ cell/mL), we observed a much higher death rate of targets than expected at the power n=0.57 (Figure A2). This indirectly supported the SiGMA model, which predicted a higher (apparent) death rate of targets at low CTL concentrations due to the non-lytically reduced tumor growth rate.

One feature of these experimental data is that the recovery of B16 tumor cells from the gels was typically lower than the desired concentration, which somewhat varied between different experiment and desired B16 cell numbers (e.g., Figure A1). Instead of fitting individual parameters to estimate the initial density of B16 tumor cells for every desired B16 concentration, we opted for an alternative approach. To predict the initial concentration of B16 tumor cells, we fit a parameter α that scaled the desired B16 concentration to the initial measured B16 concentration in the gel (see the Materials and Methods Section for more details). In separate analyses, we investigated assuming different α for different target B16 tumor concentrations by fitting our best-fit models (for Datasets 1–4 or Datasets 1–5) with one or 5 α (see Section 2 for more details and Table A2). Interestingly, the SiGMA model with varying α fit the data (Datasets 1–4) marginally better than the model with one α (F-test for nested models, p=0.02, Table A2). Other parameters such as the B16 tumor growth rate and CTL kill/suppression rates, however, were similar in both fits (Table A2). In contrast, the fits of the Sat model to all data (Datasets 1–5) were similar whether we assumed different or the same α for different target B16 tumor concentrations (p=0.37, Table A2). Because, in all cases, other statistical features of the model fit (e.g., residuals) were similar, in most of the following analyses, we considered a single parameter α in fitting the models to the data.

In our datasets, we had in total 13 gels that did not contain any B16 tumor cells after co-incubation with CTLs (Figure A1B,C); these data were excluded from the analyses so far. Data exclusion may generate biases, and we, therefore, investigated, instead of 0 B16 targets, assuming these measurements were at the limit of detection (**LOD**). The true limit of detection was not defined in these experiments, so we ran analyses assuming that the LOD = 2–10 cell/mL. Importantly, inclusion of these 13 gels at the LOD did not alter our main conclusion; specifically, the SiGMA model remained the best model for Datasets 1–4, and the Sat model remained the best model when we used Datasets 1–5 (results not shown).

### 3.3. The Best-Fit Model Varies with the Chosen Subset of the Data

The experimental data suggested that a CTL concentration of approximately $10^{6}\;\mathrm{cell}/\mathrm{mL}$ was critical for the removal of B16 melanoma cells [58]. Specifically, at concentrations of $E<10^{6}\;\mathrm{cell}/\mathrm{mL}$, the tumor cell concentration increased (Figure A1A–C), while at $E>10^{6}\;\mathrm{cell}/\mathrm{mL}$, the tumor cell concentration declined (Figure A1B–D). The growth and death rates of the tumors were similar when $E\approx 10^{6}\;\mathrm{cell}/\mathrm{mL}$, and interestingly, in one dataset, the B16 tumor concentration initially declined, but after 48 h, started to increase (Figure A1C). None of our current models could explain this latter pattern. To investigate if the data with a CTL concentration of $10^{6}\;\mathrm{cell}/\mathrm{mL}$ may bias the selection of the best-fit model, we fit our four alternative models to the data that excluded gels with desired B16 concentrations of $10^{5}$ and $10^{6}\;\mathrm{cell}/\mathrm{mL}$ and CTL concentrations of $10^{6}\;\mathrm{cell}/\mathrm{mL}$ from Dataset 3 and Dataset 4, respectively. Interestingly, for these subsets of data, the Power model fit the data with the best quality (based on the AIC), predicting that the death rate of B16 tumor cells scales sub-linearly (n=0.42) with the CTL concentration (Table A3). The Power model also provided the best fit if we included seven additional gels from Dataset 5 (with the highest CTL concentrations, Table A3) to this subset of data. Interestingly, the MA model fit this data subset with much better quality visually, even though statistically, the fit was still the worst out of all four models tested (Table A3; results not shown).

We further investigated if focusing on smaller subsets of data may also result in other models fitting such data the best. For example, in one approach, we focused on fitting the models to subsets of data with a single target B16 tumor cell concentration (Table A4). Interestingly, for B16 concentrations of $10^{4}$ and $10^{6}\;\mathrm{cell}/\mathrm{mL}$, the Power model provided the best fit, but for target B16 concentration of $10^{5}\;\mathrm{cell}/\mathrm{mL}$, the Power and SiGMA models gave the best fits. Including Dataset 5 in these analyses often led to the Sat model being the best (Table A4). Finally, dividing the data into subsets for different experiments (out of three), the Power model fit the data from Experiment 1 and 3 best, and the SiGMA model fit the data from Experiment 2 best (see Table A5). Taken together, these analyses strongly suggested that selecting the best model describing the dynamics of B16 tumor cells depends on the specific subset of data chosen for the analysis.

### 3.4. Alternative Models Predict Different CTL Concentrations Needed to Eliminate the Tumor

Given the difficulty of accurately determining the exact model for B16 tumor growth and its control by CTLs, one could wonder why we need to do that. To address this potential criticism, we calculated the time (Equation (Equation 7)) it would take for CTLs to eliminate most (90%) of the tumor cells if CTLs control tumor growth in accord with one of the four alternative models (e.g., with the parameters given in Table 1). Interestingly, the MA model predicted the largest CTL concentration that would be required to eliminate most of the tumor cells in 100 days, while the SiGMA model required the fewest ($1.54\times 10^{6}\;\mathrm{cell}/\mathrm{mL}$ vs. $0.41\times 10^{6}\;\mathrm{cell}/\mathrm{mL}$, respectively, Figure 3). The four-fold difference may be clinically substantial in cancer therapies using adoptively transferred T cells (e.g., in tumor-infiltrating-lymphocyte-based therapies [68]). Interestingly, however, the difference in the predicted CTL concentration was somewhat similar for the SiGMA and Power models, which provided the best fits for the subsets of the data (Figure 3). The range of CTL concentrations was wider between the alternative models fit to the subsets of the data (results not shown), further highlighting the need for a better, more-rigorous understanding of how CTLs control tumor’s growth in collagen–fibrin gels.

### 3.5. Mathematical Models Different from Simple Exponential Growth Are Needed to Explain B16 Tumor Dynamics in the Absence of CTLs

In our analyses, so far, we have focused on different ways CTLs can control the growth of B16 tumor cells, assuming that, in the absence of CTLs, tumors grow exponentially (Equations (Equation 3)–(Equation 6)). In our new Dataset 4, in which the gels were sampled at 0, 4, 8, 12, and 24 h after inoculation, we noticed that B16 tumor cells did not grow exponentially early after inoculation into the gels (Figure A1D). We, therefore, investigated whether a simple model in which B16 tumor cells grow exponentially is, in fact, consistent with these data.

First, we fit the exponential growth model (Equation (Equation 3) with E=0) to all data from Datasets 1–5. Interestingly, while the model appeared to fit the data well (Figure 4A) and, statistically, the fit was reasonable (e.g., residuals normally distributed), the model fits did not describe all the data accurately. In particular, the model over-predicted the concentration of B16 tumor cells at low ($10^{3}$–$10^{4}$ cell/mL) and high ($10^{8}\;\mathrm{cell}/\mathrm{mL}$) desired B16 concentrations. The lack of fit test also indicated that the model did not fit the data well ($F_{20,154}=7.12$, p<0.001). Finally, allowing the tumor growth rate to vary with the desired B16 concentration resulted in a significantly improved fit ($F_{4,170}=19.77$, p<0.001), suggesting that the growth rate of B16 tumor cells in the absence of CTLs may be density-dependent ($r_{0}=0.59/\mathrm{day}$, $r_{0}=0.65/\mathrm{day}$, $r_{0}=0.64/\mathrm{day}$, and $r_{0}=0.85/\mathrm{day}$, $r_{0}=-0.15/\mathrm{day}$ for desired B16 tumor cell concentrations of $10^{3}$, $10^{4}$, $10^{5}$, $10^{6}$, and $10^{7}\;\mathrm{cell}/\mathrm{mL}$, respectively, and α=2.48).

Second, we noticed that, in our new dataset with the B16 tumor growth kinetics recorded in the first 24 h after inoculation into the gels (Dataset 4), this did not follow a simple exponential increase (Figure A1D). Instead, there was an appreciable decline and then increase in the B16 cell concentration. We, therefore, fit an exponential growth (**EG**) model along with two alternative models, which allowed for non-monotonic dynamics—(i) a phenomenological model (**Alt1**, Equation (Equation 8)) and (ii) a mechanistic model allowing for two sub-populations of tumor cells, one dying and another growing over time (**Alt2**, Equation (Equation 9)). Interestingly, while the EG model did not fit the data well, both of the alternative models described the data relatively well (Figure 4). These analyses, thus, strongly suggested that the dynamics of B16 tumor cells in collagen–fibrin gels in the absence of CTLs are not consistent with a simple exponential growth model, at least in short-term (24 h) experiments.

### 3.6. Experiments with Several Measurements of B16 Tumor Concentrations at Variable CTL Densities Will Best Allow Discriminating between Alternative Models

In several alternative analyses, we found that the best model describing the dynamics of B16 tumor cells in collagen–fibrin gels depends on the specific dataset chosen for the analyses. It is unclear why this may be the case. One potential explanation is that individual datasets are not balanced, some have more measurements, but on a shorter time scale, while others are of a longer duration with fewer replicates. Because the exact mechanism of how CTLs impact tumor dynamics is important in predicting the concentration of CTLs needed for tumor elimination (Figure 3), we next sought to determine whether specific experimental designs may be better-suited to discriminate between alternative models [63]. We, therefore, performed stochastic simulations to generate “synthetic” data from a given assumed model for different experimental designs and tested whether, by fitting alternative models to the synthetic data, we can recover the model used to generate the data.

We considered three different designs and compared two types within each design:**Design D1**: Two-time-point experiment (Type A) vs. four-time-point experiment (Type B). The two-time-point experiments had 48 observations. The desired B16 concentrations were $10^{3}$, $10^{4}$, $10^{5}$, $10^{6}$, $10^{7}$, and $10^{8}$ cell/mL; the OT1 concentrations were 0, $10^{5}$, $10^{6}$, and $10^{7}$ cell/mL, the time points were 0 and 24 h. The four-time-point experiments had 48 observations. The desired B16 concentrations are $10^{5}$, $10^{6}$, and $10^{7}$ cell/mL; the OT1 concentrations were 0, $10^{5}$, $10^{6}$, and $10^{7}$ cell/mL; the time points were 0, 24, 48, and 72 h.**Design D2**: Short-term experiment (Type A) vs. long-term experiment (Type B). The short-term experiments had 48 observations. The desired B16 concentrations were $10^{5},10^{6}$, and $10^{7}$; the OT1 concentrations were $0,10^{5},10^{6}$,and $10^{7}$ cell/mL; the time points were 0, 8, 16, and 24 h. The long-term experiments had 48 observations. The desired B16 concentrations were $10^{5},10^{6}$, and $10^{7}$ cell/mL; the OT1 concentrations were $0,10^{5},10^{6},10^{7}$ cell/mL; the time points were 0,24,48, and 72 h.**Design D3**: More-frequent OT1 experiment (Type A) vs. less-frequent OT1 experiment (Type B). The more-frequent OT1 experiments had 40 observations. The desired B16 concentrations were $10^{5}$ and $10^{6}$ cell/mL; the OT1 concentrations were 0, $5\times 10^{5}$, $10^{6}$, $5\times 10^{6}$, and $10^{7}$ cell/mL; the time points were 0, 24, 48, and 72 h. The less-frequent OT1 experiments had 40 observations. The desired B16 concentrations were $10^{5}$ and $10^{6}$ cell/mL; the OT1 concentrations were $0,10^{4},10^{5},10^{6}$, and $10^{7}$ cell/mL; the time points were 0,24,48, and 72 h.

To draw a statistical comparison between Types A and B of the experimental designs described above, we first chose one of the Sat, Power, or SiGMA models with their best-fit parameters (Table 1) and generated 48 observations for D1 and D2 or 40 observations for D3 for Types A and B. We excluded the MA model from these analyses as it never fit the data well. In each of the generated predictions, we added an error randomly chosen from the residuals $\left(y_{i}-{\bar{y}}_{t}\right)$, where $y_{i}$ is the observed B16 count in the data and ${\bar{y}}_{t}$ is the average of $y_{i}$ at time *t*. Next, we simulated 100 replicates of such pseudo experiments, fit the three models (Sat, Power, and SiGMA) to these 100 replicates, and computed the Akaike weights to determine the best-fit model for each replicate. Due to the randomly chosen error structure for these hypothetical experiments, we found substantial variability among these 100 replicates, where the best-fit model was often different from the model from which the identical replicates were generated. For example, generating 100 simulated datasets from the Sat model, we found that the Sat model fit these data only in 52% of cases, while the Power model fit the best 36% of the time and the SiGMA model 12% of the time (first column of the first Type A matrix of Figure A5D1).

By repeating the analysis for all three models, we generated a matrix of Akaike weights with diagonal terms being heavier than the off-diagonal terms along with a constraint that the sum of a column should always add up to one (see Figure A5). In this representation, a better experimental design among each type had heavier diagonal than off-diagonal elements. Following this rule, we see that D1 (Type B), D2 (Type B), and D3 (Type A) were the better experimental designs (Figure A5). To show that the difference between the experimental designs was statistically significant, we used a resampling approach. We defined a test statistic measure given by
(12)|ΔD|=||D(A)|−|D(B)||,
where D is the determinant of the matrix and |ΔD| is the absolute difference between two determinants. Mathematically, |ΔD| is equivalent to a difference in volume of two 3D parallelepipeds, the edges of which are the columns of a matrix. For the hypothesis testing, we defined the null hypothesis as follows: the column vectors, with the constraints that the sum of the elements must be unity, belong to the same class for both experimental designs. We performed a null distribution test and a permutation test to reject the null hypothesis and showed that the column vectors that constituted the experimental designs were significantly different.

For the null distribution test, we randomly generated Type A and Type B sets of $10^{6}$ matrices with their columns being normalized to unity. |ΔD| was then computed for the the Type-A and -B designs, which formed a universal null distribution. The *p*-value was then the number of times the ${\left|\Delta D\right|}_{\mathrm{null}}$’s were greater than the observed ${\left|\Delta D\right|}_{\mathrm{obs}}$ normalized by the total number of simulations ($10^{6}$). The *p*-values for each of the designs D1, D2, and D3 (Figure 5B) confirmed that a long time experiment with more time point observations and closely spaced CTL concentrations was a significantly better experimental design. For the permutation test, we generated three column matrices from all permutations of the six columns for each of designs D1, D2, and D3. The columns were chosen from the constructed matrices of Figure A5. Then, we randomly chose sets of two matrices for Types A and B from all the permutations of the previous step. ${\left|\Delta D\right|}_{\mathrm{per}}$ was computed for Types A and B, which formed a distribution. The *p*-value was then the number of times the permuted ${\left|\Delta D\right|}_{\mathrm{per}}$’s were greater than the observed ${\left|\Delta D\right|}_{\mathrm{obs}}$ normalized by the total number of permuted sets (Figure A5). With a permutation test, we found that a long time experiment with more time point observations was a significantly better experiment, but failed to confirm the same for closely spaced OT1 concentrations with statistical significance (see the right panels of Figure A5 for *p*-values). Taken together, these simulations suggested that longer experiments (72 h) with at least four time points and a variable CTL concentration should provide the best statistical power to discriminate between alternative models of B16 tumor cell control.

## 4. Discussion

Quantitative details of how CTLs kill their targets in vivo remain poorly understood. Here, we analyzed unique data on the dynamics of SIINFEKL peptide-pulsed B16 melanoma tumor cells in collagen–fibrin gels—which may better represent in vivo tissue environments—in the presence of defined numbers of SIINFEKL-specific CTLs (OT1 CD8 T cells) [58]. We assumed that the CTL concentration is time independent in our models. This was based on one of the results of Budhu et al. [58], which showed that the activated CTL concentration >90% remained constant for at least 72 h in collagen–fibrin gels. We found that a previously proposed model in which tumors grow exponentially and are killed by CTLs proportional to the density of CTLs (mass–action law) did not describe the experimental data well. In contrast, the model in which CTLs suppress the rate of tumor replication and kill the tumors in accord with the mass–action law fit a subset of the data (Datasets 1–4 with physiologically relevant CTL concentrations of $E\leq 10^{7}$ cell/mL) with the best quality (Table 1). This result raises an interesting hypothesis that the control of tumors by CTLs may extend beyond direct cytotoxicity, e.g., by the secretion of cytokines. In fact, previous observations suggested that IFNg and TNF may suppress tumor growth in different conditions, although the ultimate effect of these cytokines on tumor progression in vivo is context-dependent as IFNg may, in fact, improve the metastasis of some tumors [65,66,67].

Importantly, however, fitting the alternative models to different subsets of data resulted in different best-fit models, e.g., including the dataset with high CTL concentrations ($E\leq 10^{8}\;\mathrm{cell}/\mathrm{mL}$) typically selected the Sat model as the best predicting that the death rate of B16 tumor cells saturates at high CTL concentrations (Table A1). In other cases, a power model in which the death rate of tumors scales sublinearly with the CTL concentration described the subsets of the data best (Table A3). The analysis of a new dataset on B16 tumor growth in the first 24 h after inoculation into gels with no CTLs suggested that a simple exponential model did not describe these data adequately; instead, models that allow for initial loss and then rebound in the concentration of B16 tumor cells were the best (Figure 4B). We also developed a novel methodology and proposed the designs of experiments that may allow better discriminating between alternative mathematical models. Our analysis suggested that longer-term experiments (0–72 h) with four measurements of the B16 cell concentration with several OT1 concentrations would have the highest statistical power (Figure 5).

Determining the exact mechanism by which CTLs control the growth of B16 tumors may go beyond academic interest. In T-cell-based therapies for the treatment of cancer, knowing the number of T cells required for tumor control and elimination is important. Our analysis suggested that the specific details of the killing term impact the minimal CTL concentration needed to reduce the tumor size within a defined time period (Figure 3). Other parameters characterizing the impact of CTLs on tumor growth may also be important (Figure 6). For example, our analysis suggested that the tumor’s growth rate, the per capita killing rate by CTLs, or the overall death rate of the tumors depends differently on the CTL concentration given the underlying model (Figure 6A–C). The latter parameter, the death rate of CTL targets, has been estimated in several previous in vivo studies (reviewed in [43]) and ranges from 0.02/day to 500/day [36,38,39,42,69,70,71,72,73]. While our estimates were consistent with this extremely broad range, whether killing of B16 tumor cells in collagen–fibrin gels occurs similarly to the elimination of targets in vivo (peptide-pulsed or virus-infected cells) remains to be determined. Interestingly, our models predicted a highly variable number of B16 tumor cells killed per day, especially at low CTL concentrations (Figure 6D). We estimated that a relatively small number of targets are killed per CTL per day, which is in line with previous estimates for the in vivo killing of peptide-pulsed targets by effector or memory CD8 T cells ([41], Figure 6D).

Our work has several limitations. First, the specifics of tumor cell and CTL movements in the gels remain poorly defined. Previous studies suggested that CTL motility in collagen–fibrin gels may be anisotropic, creating a bias in how different CTLs locate their targets [61]. Second, errors in estimating the number of surviving B16 tumor cells have not been quantified. For example, in some cases, zero B16 cells were isolated from the gels, while other gels in the same conditions contained tens-to-hundreds of cells (Figure A1B,C). In our experience, the clonogenic assays typically do not allow recovering 100% of inoculated cells, which is also indicated by the estimated parameter α>1. In fact, α=2.8 suggests that only 1/2.8=35% of inoculated B16 tumor cells are typically recovered. In the way we fit the models to the data (by log-transforming the model predictions and the data), we had to exclude gels with zero B16 tumor cells from the analysis. While this exclusion did not impact our overall conclusions, future studies may need to use methods to include zero values (i.e., censored data) in the analysis. Third, the density of the gels may change over the course of the experiment, reducing the ability of CTLs to find their targets. Using microscopy to track tumor cells and CTLs may better define if the movement patterns of the cells change over time in the gel. Fourth, the dynamics of CTLs and loss of peptides by B6 tumor cells were not accurately measured. In particular, we observed that, at a CTL concentration of $10^{6}\;\mathrm{cell}/\mathrm{mL}$ and a desired B16 tumor cell concentration of $10^{6}\;\mathrm{cell}/\mathrm{mL}$, after the initial decline, the B16 tumor cell concentration rebounded (Figure A1C). A decline in the CTL concentration with time could be one explanation; however, in other conditions, the B16 tumor cells continued declining exponentially, arguing against a loss of CTLs in the gels. The loss of recognition of tumors by CTLs (i.e., tumor escape) could be another explanation. Future experiments would benefit from also measuring the CTL concentration in the gel, along with B6 tumor cells, especially in longer (48–72 h) experiments. Fifth, the final fits of the models to the data did not pass the assumption of normality as the residuals were typically non-normally distributed (e.g., by the Shapiro–Wilk normality test). We tried several methods to normalize the residuals (e.g., excluding the outliers, using arcsin(sqrt) transformation), but none worked. Whether non-normal residuals led to biased parameter estimates of our best-fit models remains to be determined. Sixth and finally, we assumed that every CTL is capable of killing and every target is susceptible to CTL-mediated killing, which may not be accurate. Indeed, the result where the Power model fit several subsets of data with the best quality and predicted a sublinear (n<1) increase in the death rate of targets with the CTL concentration may be due to heterogeneity in the CTL killing efficacy. However, such a model would need to assume that the inoculation of CTLs into gels results in a bias of inoculating a smaller fraction of killer T cells at higher CTL concentrations, which seems unlikely.

Our work opens up avenues for future research. One curious observation of Budhu et al. [58] is that the death rate of B16 tumor cells does not depend on the concentration of the targets in the gel. We confirmed this observation as the models that included the dependence of the B16 tumor cell death rate on tumor cell concentration (e.g., the updated SiGMA model with $f_{k}=kE/\left(1+a_{1}T+a_{2}E\right)$) did not improve the fit quality, e.g., in the best fits of Datasets 1–4, we found $a_{1}\to 0$ and $a_{2}\to 0$). This model-driven experimental observation is inconsistent with the effector-to-target ratio dependence in in vitro chromium release assays and with many theoretical arguments suggesting that the killing of targets (or interactions between predators and preys) should be ratio-dependent, not density-dependent [28,44,74,75]. Interestingly, our analysis of the data from the experiments on the killing of peptide-pulsed targets in murine spleens by activated and memory CD8 T cells also showed no dependence on target cell concentration [41]. Future studies need to reconcile the difference between theoretical arguments and some in vitro experiments vs. experimental observations in gels and in vivo.

## 5. Conclusions

The hypothesis that CTLs may impact the rate of tumor growth in collagen–fibrin gels can be tested experimentally. One such experiment could be to use two populations of tumors expressing different antigens, e.g., SIINFEKL and Pmel, in the presence or absence of SIINFEKL-specific CTLs (OT-1 T cells) [76]. If CTLs reduce the growth rate of the tumor, we should detect a reduction of Pmel-pulsed tumors in gels with SIINFEKL-pulsed tumors and SIINFEKL-specific CTLs. Our experiments and mathematical-modeling-based analyses can be extended to other types of tumor cells, CTL specificity, and the type of gel. Whether the CTL killing rates estimated from in vitro data correlate with CTL efficacy in vivo remains to be determined. Effective cancer immunotherapy relies on the infiltration and killing response of CD8 T cells [77,78]. The increase of intratumoral CD8 T cells has been shown to have a direct correlation with the radiographic reduction in tumor size in patients responding to treatment [79]. In B16 preclinical melanoma models, cancer vaccines are found to induce cancer-specific CD8 T cells into tumors, leading to cytotoxicity [80]. Estimating CTL killing efficiency such as kill rate per day or the number of melanoma cells killed per day could be useful in providing guidelines on cancer immunotherapy research and, thus, our modeling platform could, therefore, provide valuable insights for estimating the efficacy of T-cell-based immunotherapies against cancer. The collagen–fibrin platform could be also useful to determine the killing efficiency of T cells (either expanded tumor infiltrating lymphocytes (**TILs**) or chimeric antigen receptor (**CAR**) T cells) prior to adoptively transferring them into patients; correlating this killing efficacy metric with the actual success or failure of the therapy in patients may be a less-expensive way to predict the overall efficacy of the therapy, thus saving time and resources.

## Figures and Tables

**Figure 1 viruses-15-01454-f001:**
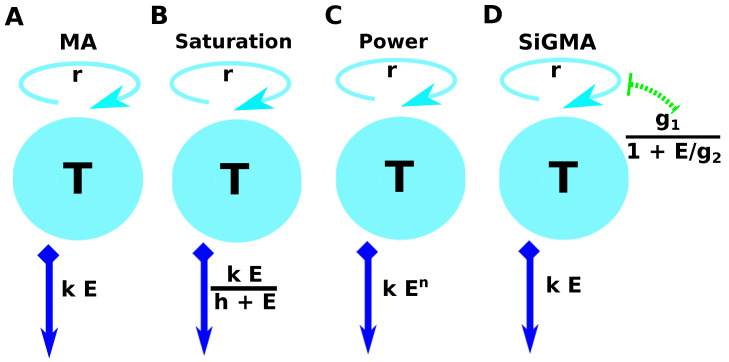
**A schematic representation of the four main alternative models fit to data on the dynamics of B16 tumor cells.** These models are as follows: (**A**) an exponential growth of tumors and a mass–action killing by CTLs (**MA**) Model (Equation (Equation 3)); (**B**) an exponential growth of tumors and saturation in killing by CTLs (**Saturation** or **Sat**) Model (Equation (Equation 4)); (**C**) an exponential growth of tumors and killing by CTLs in accord with a power law (**Power**) Model (Equation (Equation 5)); and (**D**) an exponential growth of tumors with CTL-dependent suppression of the growth and mass–action killing of tumors by CTLs (**SiGMA**) Model (Equation (Equation 6)). The tumor growth rate *r* is shown on the top of the cyan discs, which represent the B16 tumor cells *T*. For the suppression-in-growth model with a mass–action term in killing ((**D**), SiGMA), the *E* dependent suppression rate is presented over the green arrow. The killing rate *k* for each model is shown in the blue arrow pointing downwards. For example, the Power model is shown by a constant growth rate *r* with the death rate of the tumors due to *E* CTLs being $kE^{n}$.

**Figure 2 viruses-15-01454-f002:**
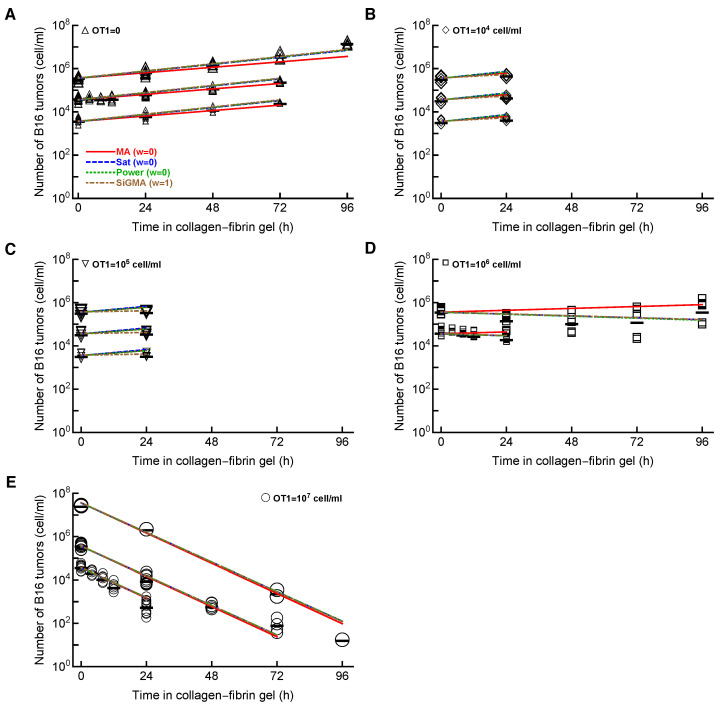
**The model assuming exponential growth of B16 tumor cells and mass–action killing by CTLs is not consistent with the B16 tumor cell dynamics.** We fit the mass–action killing (**MA**, Equation (Equation 3) and Figure 1A), saturated killing (**Sat**, Equation (Equation 4) and Figure 1B), power law killing (**Power**, Equation (Equation 5) and Figure 1C), and saturation-in-growth and mass–action killing (**SiGMA**, Equation (Equation 6) and Figure 1D) models to the data (Datasets 1–4, 431 gels), which included all our available data with CTL densities of ≤$10^{7}$ cell/mL (see Section 2 for more details). The data are shown by markers, and the lines are the predictions of the models. We show the model fits for the data for (**A**) OT1=0, (**B**) $OT1=10^{4}\mathrm{cell}/\mathrm{mL}$, (**C**) $OT1=10^{5}\mathrm{cell}/\mathrm{mL}$, (**D**) $OT1=10^{6}\mathrm{cell}/\mathrm{mL}$, and (**E**) $OT1=10^{7}\mathrm{cell}/\mathrm{mL}$. Parameters of the best-fit models and measures of relative model fit quality are given in Table 1; Akaike weights *w* for the model fits are shown in Panel A. The size of the markers denotes different desired B16 concentrations ($10^{4}$–$10^{8}$ cell/mL).

**Figure 3 viruses-15-01454-f003:**
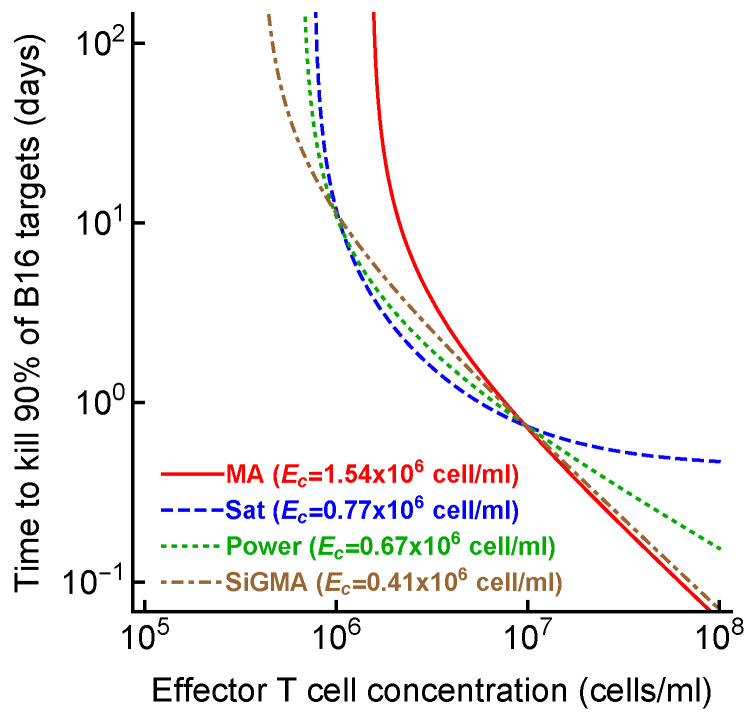
**The CTL concentration needed to eliminate most B16 tumor cells depends on the model of tumor control by CTLs**. For every best-fit model (Table 1), we calculated the time to kill 90% of B16 targets for a given concentration of CTLs (Equation (Equation 7)) from the best-fit parameters. For every model, we also calculated the control CTL concentration ($E_{c}$) that was required to eliminate at least 90% of the tumor cells within 100 days.

**Figure 4 viruses-15-01454-f004:**
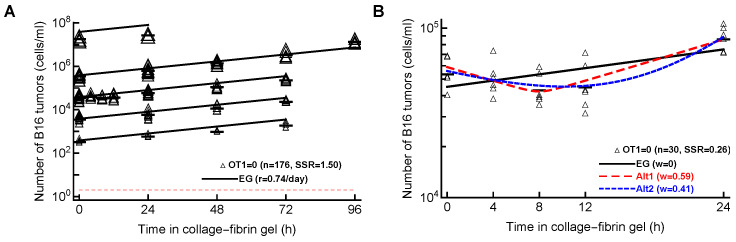
**Pure exponential growth (EG) model is not consistent with the data on B16 tumor dynamics in the absence of CTLs**. (**A**) We fit with an exponential growth model (Equation (Equation 3) with E=0) to data on B16 growth from all Datasets 1–5 with OT1=0. The best-fit values for the parameters along with the 95% confidence intervals are: α=2.6(2.4–2.8) and r=0.74(0.69–0.79)/day. (**B**) We fit exponential growth and two alternative models (Equation (Equation 3) with E=0 and Equations (Equation 8) and (Equation 9)) to the data from Dataset 4 for which OT1=0. The relative quality of the model fits is shown by Akaike weights *w* (see Table A6 for model parameters and other fit quality metrics). The data are shown by markers, and model predictions are shown by lines.

**Figure 5 viruses-15-01454-f005:**
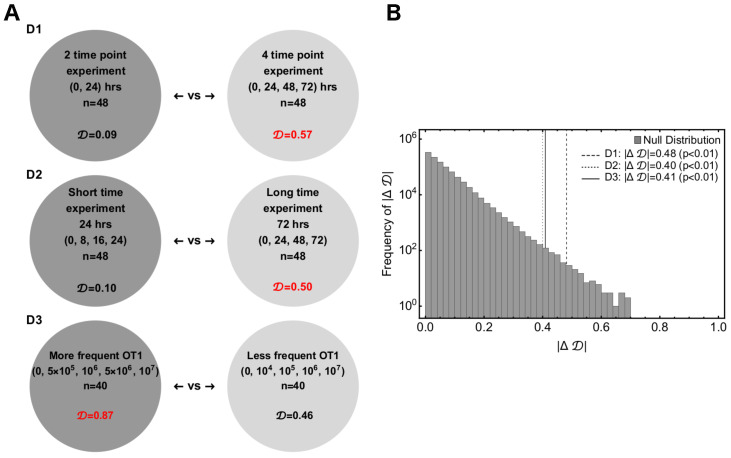
**Power analysis indicated that longer experiments with several, closely spaced CTL concentrations would allow finding the best discriminate between alternative models.** We performed three sets of simulations to obtain insights into a hypothetical future experiment, which may allow better discriminating between alternative mathematical models. (**A**) The three experimental designs are: D1—2 time-point vs. 4 time-point experiments; D2—short time scale (0–24 h) vs. long time scale (0–72 h) experiments; D3—more-frequently chosen values of CTL concentrations vs. less-frequently chosen values of CTL concentrations (see Figure A5 and Section 2 for more details). For every experimental setup, we calculated D—the determinant of a matrix formed from a simulated experimental set whose columns are constrained. (**B**) We defined a test measure ${\left|\Delta D\right|}_{\mathrm{obs}}$ between two sets of each of D1, D2, and D3 and compared the observed ${\left|\Delta D\right|}_{\mathrm{obs}}$ with the universal null distribution of ${\left|\Delta D\right|}_{\mathrm{null}}$ to compute the *p*-value. The values of D in red in Panel A show the better experimental designs in the pairs.

**Figure 6 viruses-15-01454-f006:**
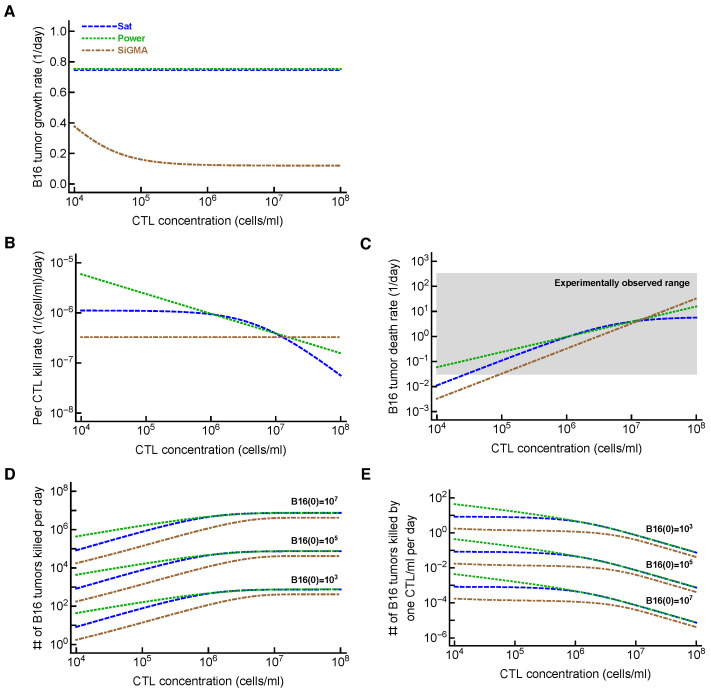
**Metrics to quantify the efficacy of CTL-mediated control of tumors are model-dependent**. For the three alternative models (Sat, Power, and SiGMA) that fit some subsets of the data with the best quality, we calculated metrics that could be used to quantify the impact of CTLs on tumor growth depending on the concentration of tumor-specific CTLs. These metrics included: (**A**) the growth rate of the tumors ($f_{g}$ in Equation (Equation 1)); (**B**) the per capita kill rate of tumors (per 1 CTL per day, $f_{k}/E$ in Equation (Equation 1)); (**C**) the death rate of tumors due to CTL killing ($f_{k}$ in Equation (Equation 1)); the grey box shows the range of the experimentally observed death rates of targets as observed in some previous experiments (see Section 4 for more details and [43]); (**E**) the total number of tumors killed per day as a function of 3 different initial tumor cell concentrations (indicated in the panel); (**D**) the number of tumors killed per 1 CTL/mL per day. The latter two metrics were computed by taking the difference of the growth and combined killing at 24 h. The parameters for the models are given in Table 1, and the model equations are given in Equations (Equation 4)–(Equation 6).

**Table 1 viruses-15-01454-t001:** **Parameters of the 4 alternative models fit to Datasets 1–4 (excluding data with $\mathrm{CTL}=10^{8}\;\mathrm{cell}/\mathrm{mL}$) and metrics of relative quality of the model fits**. Estimated parameters: α is a dimensionless inverse of the effective fraction; *r* is given in units of per day; *h* is in cell/mL; $g_{1}$ is in per day; $g_{2}$ is in cell/mL. The parameter *k* had different units in different models: per OT1 cell/mL per day, per day, per OT1 (cell/mL)${}^{n}$ per day, and per OT1 cell/mL per day for MA (Equation (Equation 3)), Sat (Equation (Equation 4)), Power (Equation (Equation 5)), and SiGMA (Equation (Equation 6)), respectively; *n* is a dimensionless parameter. The parameter estimates and 95% confidence intervals for the best-fit SiGMA model were: α=2.71(2.5–2.9), $g_{0}=0.12\;$(0.036–0.2)/day, $g_{1}=0.64\;$(0.55–0.73)/day, $g_{2}=6.72\;$(4.14–17.57) $\;\times \;10^{3}\mathrm{cell}/\mathrm{mL}$, and k=3.3(3.15–3.4)$\;\times \;10^{-7}\mathrm{cell}/\mathrm{mL}/\mathrm{day}$; the model fits are shown in Figure 2. The best-fit model (with the highest *w*) is highlighted in blue.

Datasets 1–4 ($E\leq 10^{7}$ cell/mL): *n* = 431												
**Model**	α	*r*, **1/day**	*k*	*h*	*n*	$g_{0}$	$g_{1}$	$g_{2}$	**SSR**	**AIC**	Δ	*w*
**MA**	2.77	0.576	$3.79\times 10^{-7}$						164	814	160	0
**Sat**	2.81	0.744	6	$5.34\times 10^{6}$					118.5	677	23	0
**Power**	2.79	0.744	$2.23\times 10^{-4}$		0.606				116	668	14	0
**SiGMA**	2.71		$3.29\times 10^{-7}$			0.12	0.64	6715	112	654	0	1

## Data Availability

The data for the analyses are provided as a Appendix A to this publication and on github: github.com/vganusov/killing_in_gels (accessed on 6 June 2023).

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
