# Peer review of "Cytotoxic T Lymphocytes Control Growth of B16 Tumor Cells in Collagen–Fibrin Gels by Cytolytic and Non-Lytic Mechanisms"

_viruses, 2023, doi:10.3390/v15071454_

Round 1

Reviewer 1 Report (Previous Reviewer 1)

The authors have addressed most of my previous comments. Kudos to them for that.

One comment, however, remains outstanding: Fig 2 does not show visually what is claimed.  The obvious reason is that the authors lump together 4 datasets, in different orders of magnitude,, and the fine difference in fitting is impossible to resolve (compare Fig 4A and 4B).

The only valid conclusion from Fig 2 and Fig 4A is that the fitting of all 4 datasets at once not make it possible to differentiate between the models; they have to be used separately, as in Fig. 4B, which looks convincing.

Table 1 does not  help either, because its columns have no titles: what is shown in them, remains a mystery, and the legend does not help.

Hence, I have to double down on my previous suggestion, that the authors hide Fig 2 and the text in Supplement, and make another, more striking figure the opening of the Results. For example, Fig. 4.

Because the authors have a very interesting text later on, e.g., on experiment design, I do not wish to see the paper starting from a disappointing figure with unsupported claims.

Author Response

Response to reviewers’ comments to the revised version of the paper.

Reviewer 1.

One comment, however, remains outstanding: Fig 2 does not show visually what is claimed.  The obvious reason is that the authors lump together 4 datasets, in different orders of magnitude,, and the fine difference in fitting is impossible to resolve (compare Fig 4A and 4B).

Response. We again, respectfully, disagree. Combining all the data into one large dataset for the analysis is important as we showed that looking at specific subsets of the data results in different models to be selected as best (based on the model quality of the fit judged by AIC). We fully agree that top three models fit the data reasonable well, so the difference indicated by AIC differences (or Akaike weights) is not visible in the graphs. However, the worst fit model, the MA model, clearly does not fit the data well – see, for example, Fig 2A and 2D. Also remember that AIC is proportional to the log-likelihood of the model, so a difference of 10 in AIC is very large (proportional to exp(10)!) as has been discussed previously (see book by Burnham and Anderson 2002).

The only valid conclusion from Fig 2 and Fig 4A is that the fitting of all 4 datasets at once not make it possible to differentiate between the models; they have to be used separately, as in Fig. 4B, which looks convincing.

Response. We agree that visually it is not possible to tell which of the 3 models (not 4, only 3) fit the data best. We have to involve statistics for that. In a way, we agree that a difference of AIC of 14 between the SiGMA and Power models may not seem large but statistically it is. This is why we actually state that future experiments need to be done to discriminate between the models more rigorously. But given the current data, SiGMA model fits all the data with best quality based on AIC – and judging model fit quality on AIC with larger (>10) difference is an acceptable practice in modeling (see book by Burnham and Anderson 2002). 

Table 1 does not  help either, because its columns have no titles: what is shown in them, remains a mystery, and the legend does not help.

Response. The reviewer must have looked at the version with tracked changes which indeed did not have column headers. We apologize for that. In producing this version using latexdiff script, top row generated errors and was removed. However, the original version (with no changes highlighted) the headers were present and clearly indicated statistical advantage of the SiGMA model – which again, we agree, may not seem big but it is based on statistical properties of AIC. Perhaps in the new version of the paper, formatted to Viruses template, the reviewer could now agree that this model fits the full dataset statistically better.

Hence, I have to double down on my previous suggestion, that the authors hide Fig 2 and the text in Supplement, and make another, more striking figure the opening of the Results. For example, Fig. 4.

Response. We hope with new formatting of the paper and clear statement by Table 1, the reviewer would accept our decision to keep main Figure 2 in the text as it is very important for the paper.

Because the authors have a very interesting text later on, e.g., on experiment design, I do not wish to see the paper starting from a disappointing figure with unsupported claims.

Response. Well, we don’t believe that our figure is disappointing. Let’s readers decide on that, we are happy to take the blame if this will be the decision of the community.

Reviewer 2 Report (Previous Reviewer 2)

I have gone through the revised paper carefully and can confirm that the authors have significantly addressed all my queries and suggestions. Tehrefore, the paper can be accepted for publication. 

Author Response

No need. Reviewer agreed with our revisions.

This manuscript is a resubmission of an earlier submission. The following is a list of the peer review reports and author responses from that submission.

Round 1

Reviewer 1 Report

The authors compare several alternative models of killing and growth suppression of cancer cells by CTLs that differ in the dependence of the cell killing and virus suppression rates on CTL concentration with published and unpublished data obtained in a gel-based assay. They show that the simplest mass-action model, generally, does not work, and that the best-fit model varies between datasets. They propose new experimental setups that would better differentiate between the models, and explain why knowing the best-fit model is important therapeutically.

I think that this work is biomedically sound and educating, but could benefit from some reorganization and rewriting.

Main comments:

Lines 77-78: It is unclear how the killing time was estimated in the first case, when it is compared with the second case.

Line 80: May be, “endogenous antigen” instead of “exogenous”? CTLs, in the in vivo conditions recognize antigen produced inside of a cell. Also, peptide pulsing is also exogenous, so the phrase looks odd.

Line 161 and equation 2: The authors assume that the inoculated CTLs did not change their numbers during the experiments. CTLs can be activated by the pre-pulsed target cells and undergo both division and death. The evidence for this assumption has to be stated briefly and discussed in Discussion.  How sensitive are the conclusions to this assumption?

Line 163. The mathematical term “scaling factor” is used incorrectly. Perhaps, parameter 1/\alpha is “the efficacy of inoculation”? “Effective fraction of cells”?

Lines 126, 130, 135, 146, 162, 164, 187 and some others: The term “target concentration” is a bad choice, because it associates with the pre-pulsed target cells. I would use “desired concentration” or “planned concentration”.

Line 253: The section title is misleading because this result is not seen until much later, in Fig 4. The correct title is “Fitting alternative models” or something like this.

Lines 272-287 and Figure 2. These results do not show that MA law does not fit data well. Only Figure 4B does. The differences in Akaike criterion are too small and should not be relied on to derive conclusions.

Fig 2. This figure is uninformative and should be hidden in Supplement. The text related to it should be shorten a lot, and the existing text go to Supplement as well. Also, I can see only red and grey color in the figure. Where are the other colors promised in the legend?

Fig S1 and S2: Instead of the boring Fig 2, the interesting Figs S1 and S2 should go to the main text. They are discussed a lot and are important for authors’ statements in the Abstract.

Fig 3: Are those best-fit parameters, or they are chosen to have all lines intersect at one point E=10^7, and three lines intersect at E=10^6? A comment on these two fixed points would help.

Line 326-352: This is the most educating result of the paper and should come earlier.

Fig 4B is the first figure that obviously shows that MA law does not fit well. It matches nicely with the title in Line 253. Can you show it earlier to back your claim in the Abstract?

Lines 407-423: Perhaps, experiments should also be longer than 72 hours if only possible. A week?

Minor comments:

Introduction and Methods can benefit from copy editing for better grammar and style. A sample follows.

Line 37: replace “some” to “many” or drop completely.

Line 38: change to “including direct cytotoxicity and...” and drop “is important”

Line 56: Only one study?! I thought MA is naturally expected in the dilute limit.

Line 62: Drop “one”

Line 64: the target cells transfer

Line 68: a mass action killing term

Lines 69,75, 77: comma after [42], “that”, and no comma after “cells” and “parasites”, respectively.

line 77: hours, which is longer

Line 82: “zombie contracts” require an explanation, quotation marks, and a separate reference. Not every reader is into zombies.

Line 87: “have a low throughput”

Line 90 and 91: “the complex”, “a low-cost”

Lines 93 and 94: no commas

Line 101: “In the present paper, we re-analyze the data by Budhu et al [] more rigorously, along with..”

Line 103: “we find out...”

Line 104. “the best-fit model, out of the four tested, is dependent on a specific dataset”

Line 107: “fit, the best”

Lines 110-11: ‘’simulate... find out which”

Line 112: “would allow”

Line 116: “The details of experimental designs are described in [1].”

Line 122: “The data represent the concentration”

Line 123: “kindly provided us”. (They did not have to do it, right?)

Line 125: “inoculated in 3D”, no “a”

Lines 126, 130, 135, 141, 146, 162: “at the desired initial concentrations” (see a comment above)

lines 130, 135, 142: “cell/ml, together with”.  Drop “each”.

line 130 and below: cell/ml not cells/ml

line 145: comma after “this”

Line 152: “...and, at a time point, each...”

Line 157: Comma after [1]

Line 164: “than it was inoculated”.

Reviewer 2 Report

I have read the paper. In my opinion, the manuscript is a candidate for publication in this journal. However, it must be revised according to the following comments:

1. The current abstract is too lengthy. It must be concise and must contain answers to the following questions: Why is this particular study important? What are the important results from this study different from others? What conclusions can be drawn from the results in this study?

2. The major assumptions of the governing equations must be supported by strong biological evidences.

3. The methdology behind the paramter estimation must be supported by references or did the authors develop his own methods for the fitting and estimation? Please be more detailed

4. Please number the main sections and subsections in the manuscript according to journal standard. Please take some time and organize this paper very neatly according to standard formats of this journal.

5. Relate the results of this study with previous works in this direction and justify how this current model improved on previous existing ones in a more scholarly way

6. Discuss the convergence, stability and accuracy of the used numerical scheme? How is it better than other schemes not used

7. Authors must avoid too many irrelevant citations such as: [37, 39, 40, 43, 68{72] in the revision. Delete as much references that are not really relevant and capture ONLY relevant papers needed in the work. 80 References is too much for this paper, from my checks. Please do some work on this.

8. There should be a Conclusion Section, which must summarise important results and findings from this study. It should also capture detailed future directions 

 9. The entire should be thoroughly checked for typos and also the typesetting should be greatly improved to attract wider readership. Make the paper more interesting to readers.

Reviewer 3 Report

In this paper, the authors presented mathematical models that made different assumptions of how CTLs impact B16 tumor cells. The models can suggest cytotoxic T lymphocytes control growth of B16 tumor cells in collagin-fibrin gels by cytolytic and non-lytic mechanisms.

I think the problem considered in this paper is interesting, and it can be accepted to appear in Viruses.